# Progress of Conjugated Linoleic Acid on Milk Fat Metabolism in Ruminants and Humans

**DOI:** 10.3390/ani13213429

**Published:** 2023-11-06

**Authors:** Kun Wang, Zimeng Xin, Zhi Chen, Huanan Li, Diming Wang, Yuan Yuan

**Affiliations:** 1Key Laboratory of Molecular Animal Nutrition, Zhejiang University, Ministry of Education, Hangzhou 310058, China; mx120210857@stu.yzu.edu.cn (K.W.); meng1217180507@163.com (Z.X.);; 2College of Animal Science and Technology, Yangzhou University, Yangzhou 225009, China; zhichen@yzu.edu.cn; 3School of Nursing, Yangzhou University, Yangzhou 225009, China

**Keywords:** fatty acids, conjugated linoleic acid, milk quality, ruminant milk fat

## Abstract

**Simple Summary:**

The nutritional composition of dairy products is regulated by many factors, and it is an important question, how to make the milk produced by ruminants better, as well as to discover the important regulatory factors involved. Conjugated linoleic acid (CLA), as one of the natural fatty acids present in ruminant dairy products, has been found by researchers to have anticancer, anti-inflammatory, immunomodulatory, and lipid metabolism regulation properties, as well as a role in infant growth and health. Meanwhile, different CLA have a modulating effect on the right milk fat percentage in ruminant dairy products, with *cis*-9, *trans*-11 CLA and *trans*-10, *cis*-12 CLA being the representative isomers. It is therefore necessary to explore the potential of CLA in improving animal performance and the nutritional value of livestock products. We believe that the study of how CLA regulates milk fat synthesis is both novel and valuable, and that there is a need to elucidate the mechanism of CLA regulation in ruminant milk fat and breast milk fat.

**Abstract:**

As a valuable nutrient in milk, fat accounts for a significant proportion of the energy requirements of ruminants and is largely responsible for determining milk quality. Fatty acids (FAs) are a pivotal component of milk fat. Conjugated linoleic acid (CLA) is one of the naturally occurring FAs prevalent in ruminant dairy products and meat. Increasing attention has been given to CLA because of its anti-cancer, anti-inflammatory, immune regulation, and lipid metabolism regulation properties, and these benefits potentially contribute to the growth and health of infants. In breast milk, CLA is present in trace amounts, mainly in the form of *cis*-9, *trans*-11 CLA. Notably, *cis*-9, *trans*-11 CLA improves the milk fat rate while *trans*-10, *cis*-12 CLA inhibits it. Apart from having multiple physiological functions, CLA is also a pivotal factor in determining the milk quality of ruminants, especially milk fat rate. In response to growing interest in green and healthy functional foods, more and more researchers are exploring the potential of CLA to improve the production performance of animals and the nutritional value of livestock products. Taken together, it is novel and worthwhile to investigate how CLA regulates milk fat synthesis. It is the purpose of this review to clarify the necessity for studying CLA in ruminant milk fat and breast milk fat.

## 1. Introduction

Dairy cows’ milk fat composition is an important indicator of quality, since milk fat is a high-value component of ruminant milk. Ruminant milk contains a wide range of bioactive fatty acids (FAs), whose content varies depending on animal varieties, physiological periods, diet nutrition, and other factors. Furthermore, FAs in ruminant milk primarily come from short-chain FAs (C4–C8) and medium-chain FAs (C9–C14). Both fatty acids are produced de novo in mammary epithelial cells, whereas long-chain fatty acids (>C16) are largely absorbed directly from the blood. Also, about half of the C16:0 is partially synthesized in the mammary glands and the other is obtained from the gut [1,2].

There has been a growing body of evidence showing that *trans*-10, *cis*-12 CLA strongly inhibits milk fat synthesis, supporting the hypothesis that the altered fat biohydrogenation pathway in the rumen leads to milk fat depression (MFD). When ruminants are fed a high-concentrate diet, MFD occurs, characterized by a significant reduction in milk fat production without affecting milk yield and milk protein production [3]. MFD begins with a decrease in the synthesis of FAs in the mammary tissue of cows, which is caused by the addition of CLA. This is now explicitly associated with the *trans*-10, *cis*-12 CLA isomer. As a result of CLA-induced MFD, genes that encode milk fat synthesis as well as key transcription factors are downregulated, providing molecular evidence for the suppressive effect of CLA on milk fat. In addition to the lactation period, there is usually a negative energy balance [4]. Dietary supplementation with *trans*-10, *cis*-12 CLA is shown to reduce milk fat synthesis and promote energy transfer from mammary glands to adipose tissue, thus ameliorating MFD to some extent.

Related studies in human medicine have established that CLA exhibits positive effects on human health. CLA exhibits numerous health benefits such as anti-obesity, anticarcinogenic, antihypertensive, antidiabetogenic, immunomodulatory, and osteosynthetic effects, besides being used for the treatment of cardiovascular disease, metabolic syndrome, and asthma. Most of these biological effects have been attributed to the *cis*-9, *trans*-11 and *trans*-10, *cis*-12 CLA isomers [5]. In addition to ruminant dairy products, *Bifidobacterium* and *Lactobacillus* colonized in the human gut have also been shown to synthesize CLA [6]. This review examines the structure of CLA, the mechanism of CLA biosynthesis in ruminants, the biological sources of CLA in breast milk, and the effects of *trans*-10, *cis*-12 CLA and *cis*-9, *trans*-11 CLA on milk fat synthesis.

## 2. Structure of CLA

CLA is a term used to describe a mixture of isomers of linoleic acid (LA) that have conjugated double bonds at different positions. Each double bond can be either in *cis* or *trans* (Figure 1). It is generally believed that its conjugate double bond begins with the 8th, 9th, 10th and 11th carbon atoms at the carboxyl end [7]. Four configurations of CLA have been documented: *cis*-*trans*, *trans*-*cis*, *cis*-*cis*, and *trans*-*trans*. CLA comprises 28 positions and geometric isomers [8], but only *cis*-9, *trans*-11 CLA and *trans*-10, *cis*-12 CLA have been demonstrated to exhibit biological activity. Generally, the former positively affects animal immunity, anticancer activity and production performance [9], while the latter primarily alters animal body fat metabolism and reduces fat deposition [10]. Additionally, *trans*-10, *cis*-12 CLA significantly influences the amount and composition of milk fat during manufacturing processes [11].

## 3. Biosynthetic Process of CLA in the Milk

The biosynthesis of CLA in ruminants involves two primary pathways. The first one relies on the biohydrogenation of substrates LA and linolenic acid (LNA) by rumen microorganisms, and the other occurs in animal tissues such as the mammary gland, where biohydrogenated intermediates are synthesized through the action of desaturases [12]. In addition, CLA has been identified in human breast milk, and bifidobacteria and lactobacilli, which colonize the maternal gut, are believed to be its significant producers in the human body.

### 3.1. Biosynthesis Process of CLA in Ruminants

#### 3.1.1. Hydrogenation of Rumen Microorganisms

Numerous experiments have demonstrated that the hydrogenation of unsaturated FAs by rumen microorganisms is primarily dependent on bacteria, especially Vibrio butyrate. The bacteria involved in biohydrogenation can be categorized into two groups, Group A (i.e., *B. fibrisolvens*) and Group B (i.e., *Fusocillus* spp., *B. proteoclasticus*). Group A bacteria isomerize and hydrogenate LA to trans-vaccenic acid (TVA), whereas Group B bacteria saturate TVA to stearic acid [13,14]. The complete biohydrogenation of unsaturated FAs necessitates a balance between these two groups. Upon digestion and absorption of dietary lipids by ruminants, the lipids initially undergo hydrolysis into LA (*cis*9, *cis*12 18:2) and LNA (*cis*9, *cis*12, *cis*15 18:3). Under a normal rumen metabolism, LA undergoes the enzymatic action of linoleic acid isomerase (LAI) in group A bacteria, resulting in the formation of a conjugated double bond structure in the *cis*-9, *cis*-12 18:2 structure of CLA. Subsequently, rapid hydrogenation leads to the formation of TVA (*trans*-11 18:1), which experiences slow hydrogenation to yield saturated fatty acid C18:0 and is then accumulated and absorbed in the rumen. Similar to the LA hydrogenation process, the LNA hydrogenation process also yields TVA as an intermediate product, with the final product being C18:0 [15]. However, the initial steps of LA biohydrogenation involve double bond isomerization at the *cis*-12 position, resulting in the formation of *cis*-9, *trans*-11, *cis*-15. Due to the sluggish rate of TVA reduction and its consequent accumulation in the rumen, TVA hydrogenation is frequently employed as a rate-limiting step in the complete hydrogenation of unsaturated FAs when different microbiomes are involved [16]. These microorganisms are responsible for the biological hydrogenation of most dietary CLA isomers to form TVA. The unhydrogenated rumen acid is absorbed in the gastrointestinal tract and subsequently transported via the bloodstream to various tissues, such as liver, mammary gland, and fat [17].

#### 3.1.2. Endogenous Synthesis of CLA

Numerous studies have shown that endogenous synthesis of *cis*-9, *trans*-11 CLA accounts for approximately 64% of milk fat CLA, making it the primary origin of milk fat CLA. Sterculic oil inhibits Δ^9^-Fatty acid desaturase (Δ^9^-desaturase) and then significantly reduces milk fat CLA, further suggesting the role of endogenous synthesis [18]. The *cis*-9, *trans*-11 CLA formed in the rumen can either be absorbed directly or metabolized by rumen microorganisms to produce TVA, which is subsequently absorbed by tissues and desaturated to recreate *cis*-9, *trans*-11 CLA in the mammary gland. This entire process necessitates the involvement of Δ^9^-desaturase. In cows and dairy goats, the mammary gland exhibits the highest Δ^9^-desaturase activity [19].

Currently, studies on the biosynthesis of CLA in the mammary gland of ruminants have focused on *cis*-9, *trans*-11 CLA (Figure 2). The production of *trans*-10, *cis*-12 CLA has only been observed in Propionibacterium seizures and Escherichia coli, yet these two strains have not been detected in ruminants [16]. In addition to the involvement of rumen bacteria in the *cis*-9, *trans*-11 CLA biohydrogenation, some other bacterial strains such as bifidobacteria, lactobacilli, and murine intestinal bacteria have been identified and isolated [20]. Among them, *Bifidobacterium* and *Lactobacillus* have been shown to produce CLA in the human body. In ruminants, however, they cannot participate in hydrogenation due to their inability to metabolize free LA or the antagonism between LA and other limitations.

### 3.2. Sources of CLA in Human Infant

Breast milk contains 7.19% *trans* polyunsaturated FAs on average, mainly in the conformation of *cis*-9, *trans*-11 CLA [21]. CLA in breast milk has been documented to modulate a variety of key processes in the pathogenesis of asthma, including PPARγ-dependent and non-dependent inflammation, arachidonoid production, and humoral immune responses, making it possible to be an effective therapy for infant asthma [22]. In addition, it strengthens the infant’s immunity and prevents the development of allergic asthma [23]. On the downside, breast milk contains only trace amounts of CLA in comparison to ruminant dairy products, particularly yak milk. Nonetheless, a lipidomic differential analysis reveals that the breast milk lipids contain a higher proportion of LA than cow and goat milk [24]. It is noteworthy that mothers rely on exogenous intake for CLA as opposed to endogenous synthesis. The consumption of CLA-rich foods by lactating mothers has been observed to result in a substantial increase in the amount of CLA in breast milk, thus fulfilling the CLA requirements of their infants [25]. In a manner akin to rumen bacteria, live lactic acid bacteria and bifidobacteria in the intestines of infants are producers of CLA, and breast milk serves as a crucial source of these two microorganisms [26,27].

#### 3.2.1. Bifidobacterium

*Bifidobacterium* is a dominant strain in the intestine of breastfed healthy newborns, with a colonization rate ranging from 60% to 90% in the intestine. It plays a pivotal role in promoting neonatal immune maturation and resisting pathogenic bacteria, therefore contributing to the early health of infants [28]. In terms of strain levels, infant intestinal bifidobacteria mainly include short-chain bifidobacteria, long-chain bifidobacteria, *Bifidobacterium* infantis, and *Bifidobacterium* bifidum [29]. Initially isolated from infant feces in 1899, short-chain *Bifidobacterium* (*B. breve*) is a genus of bacteria that is Gram-positive, nonmotile, rod-shaped, and strictly anaerobic [30]. Among them, *B. breve* NCFB 2258 exhibits a conversion efficiency of 66.2% [31], while *B. breve* WC0421 demonstrates a superior capacity to produce *cis*-9, *trans*-11 CLA, as evidenced by GC-MS analysis [32]. Additionally, *B. longum* and *B. dentium* have been validated as other species with high CLA production potential. Specifically, *Bifidobacterium longum* DPC6320 and *Bifidobacterium longum* DPC6315 exhibit conversion rates of 43.89% and 11.0%, respectively [33,34]. For *B. dentium*, its conversion rate clusters between 10% and 30%. Notably, the earliest colonization and most robust transformation rate of infant *Bifidobacterium* is merely 3.31% [35].

#### 3.2.2. *Lactobacillus*

*Lactobacillus*, a microorganism found in infant feces, is capable of synthesizing CLA under both in vitro and in vivo conditions. Prior research has indicated that lactobacilli from infant feces exhibits higher CLA production than those from adult feces, with lactobacilli from dairy products following closely behind. Among these, *Lactobacillus* plantarum, which can be isolated from breast milk, has been identified as a high CLA producer. Specifically, *Lactobacillus* plantarum ZS2058 has been shown to convert over 50% of LA into *cis*-9, *trans*-11 CLA [36,37]. Furthermore, *Lactobacillus* plantarum AKU1009a can convert up to 85% of LNA into *cis*-9, *trans*-11 CLA and *trans*-10, *cis*-12 CLA [38].

It is worth mentioning that, because milk contains a lot of oligosaccharides, the lactic acid-producing bacterial genera *Lactobacillus* and *Bifidobacterium* are usually more common in pre-weaned ruminants. The rumen’s relative abundances of the bacterial genus Bifidobacterium remain constant from birth to an age of 83 days old. The ruminal fluid of 2-day-old and 15–83-day-old calves did not contain any bacteria belonging to the genus *Lactobacillus*, but 3–15-day-old calves had low relative abundances of the bacterium [39]. And the pre-weaned calves had higher fecal relative abundances of the genera Bifidobacterium, Faecalibacterium, and *Lactobacillus* than post-weaned calves [40].

Cesarean births may benefit from breastfeeding due to its potential to improve microbiota. As evidenced by numerous experiments, more than 200 species of bacteria have been detected in breast milk, including Staphylococcus, Streptococcus, Corynebacterium, *Lactobacillus*, Propionibacterium, and Bifidobacterium, in spite of breast milk being traditionally considered sterile [41]. Endogenous contamination is the source of breast milk microorganisms. The microbiota in breast milk has a significant impact on the microbiota of the infant’s digestive tract. The feces of breast milk and breastfed infants contain identical strains of *Lactobacillus*. However, a persistent challenge is the inadequacy of dairy sources of CLA in meeting the population’s needs, thereby necessitating additional supplementation. The ability of Bifidobacterium and *Lactobacillus* in breast milk to bioconvert LA to CLA may explain the need for Bifidobacterium and *Lactobacillus* supplementation during lactation. Unfortunately, a dearth of research on the precise mechanism by which CLA is synthesized in breast milk remains a concern.

## 4. The Effect of *trans*-10, *cis*-12 CLA on Milk Fat Synthesis

The inclusion of natural-type CLA in ruminant feed and its subsequent ingestion by animals impacts milk lipid synthesis. *Trans*-10, *cis*-12 18:2 reduces milk fat content and yield by 42% and 44%, respectively, after 4 days of true gastric infusion [42]. In another study, dietary CLA supplementation increases milk production in lactating Holstein cows but decreases concentrations of milk fat and short- and medium-chain fatty acid [43]. Additionally, the anti-fat effect of *trans*-10, *cis*-12 CLA is observed in dairy cows, albeit to a lesser extent than in bovine or ovine [44]. The dose-dependent relationship between *trans*-10, *cis*-12 CLA and milk fat synthesis is curvilinear, with a maximum inhibition on milk fat secretion of approximately 50% [45].

### 4.1. Regulation of De Novo Synthesis by trans-10, cis-12 CLA

Approximately half of FAs in milk are synthesized by de novo, with the majority being short-chain FAs and some medium-chain FAs. The regulation of this process is mainly governed by Acetyl-CoA Carboxylase Alpha (ACACA), Fatty Acid Synthase (FASN), and acyl-coa synthetase short-chain family member 2 (ACSS2) (Figure 3). These three enzymes have been detected to undergo horizontal upregulation during lactation. Once entering the mammary epithelium from the bloodstream, acetic acid and β-hydroxybutyric acid are converted into acetyl coenzyme A and β-hydroxyisobutyl coenzyme A, respectively, through the action of ACSS2. Acetyl coenzyme A is then carboxylated by ACACA, leading to the formation of malonyl coenzyme A. Subsequently, FASN facilitates the sequential addition of two carbon atoms to the growth chain of FAs [46].

The endogenous synthesis of FAs in ruminant dairy products originates from the fat of consumed feed and the mobilization of body fat, particularly in early lactation. FAs, including CLA, exhibit varying degrees of regulatory influence on this process. A number of studies have reported that CLA infusion in the abomasum of ruminants results in a reduction in the proportion of short- and medium-chain FAs in milk fat, leading to an increase in the proportion of LCFAs [47]. These findings suggest that the mechanism by which CLA reduces the synthesis of milk fat involves a potent inhibition of de novo synthesis of FAs in the mammary gland [24,44]. The proteins encoded by FASN and ACACA play a significant role in this process in bovine mammary epithelial cells. In fact, these proteins serve as target genes of CLA and contribute to the formation of a pathway that regulates milk fat synthesis. FASN expression is detected to be downregulated after CLA treatment [48]. It is postulated that the aforementioned inconsistency may be attributed to the fact that, at low concentrations, CLA does not preferentially inhibit the de novo synthesis of FAs. After administration of high doses (10 g/d) of *trans*-10, *cis*-12 CLA to cows intravenously for a period of time, a 57% reduction in FASN mRNA abundance and a 46% reduction in ACACA levels were detected [49]. In conclusion, the regulation of milk fat synthesis by CLA involves the inhibition of mammary fatty acid de novo synthesis, which can be achieved by reducing the expression of genes related to milk fat de novo synthesis.

### 4.2. Regulation of LCFAs Uptake and Transport in Milk Fat by trans-10, cis-12 CLA

It is only through the circulating blood that LCFAs in the mammary gland are actively or passively absorbed [46]. This process involves blood very-low-density lipoproteins (VDLD) and celiac-dependent low-density lipoprotein (LDL) anchoring the mammary endothelium, and subsequently LCFAs entering the mammary epithelium via protein-mediated transport. Proteins involved in this process include solute carrier 27 (SLC27), cluster of differentiation 36 (CD36), long-chain acyl cofactor A (ACSL) synthetase, fatty acid binding protein (FABP), and lipid acid transport protein (FATP). Besides, there is great activity in primary bovine mammary epithelial cells (BMEC) when it comes to synthesis, utilization, and transport of FA. A growth in transcription and upregulation of CD36 with increasing CLA concentrations can be detected in mammary epithelial cells cultured in vitro after 150 mol/L *trans*-10, *cis*-12 CLA treatment. Consistently, in mammary epithelial cells of dairy goats, *trans*-10, *cis*-12 CLA can upregulate CD36 expression while inhibiting FABP3 [50]. Because CD36′s ability to bind LCFA facilitates their entry into cells, this may explain the paradoxical ability of CLA to inhibit fatty acid de novo synthesis but promote intracytoplasmic lipid accumulation [51]. Inadequately, experimental proof is yet to be determined.

### 4.3. Regulation of Desaturation of Milk Fat by trans-10, cis-12 CLA

The unique hydrogenation of the rumen results in only a small fraction of the FAs absorbed by the mammary gland being unsaturated. Formed by ab initio synthesis, the 14-16-carbon saturated FAs are converted into monounsaturated FAs by the formation of a *cis* double bond at the Δ^9^ position catalyzed by Stearoyl-CoA desaturase (SCD) [52]. The reduced concentration of desaturation reaction products suggests that CLA blends tend to attenuate the desaturation of LCFAs, thus altering the fatty acid composition. SCD1 is the primary source of Monounsaturated FAs in milk fat. After being fed equal proportions of *trans*-10, *cis*-12 CLA and *cis*-9, *trans*-11 CLA, lower SCD enzyme activity in mammary gland tissue and higher concentration of stearic acid in milk are observed in the *trans*-10, *cis*-12 CLA treated group [53].

Therefore, it appears that *trans*-10, *cis*-12 CLA is a more effective inhibitor of mammary gland desaturation. Moreover, treatment of goat mammary epithelial cells with *trans*-10, *cis*-12 CLA results in a decrease in SCD1 abundance, which is further substantiated by the decrease in C16 and C18 desaturation indices [54]. In contrast, *cis*-9, *trans*-11 CLA exhibits significantly lower efficacy in regulating SCD. The low expression of SCD protein in lactating sheep mammary gland cells attributable to the increase of DNA methylation significantly curtails the content of *cis*-9, *trans*-11 CLA in milk. This phenomenon is predicated on the endogenous synthesis of *cis*-9, *trans*-11 CLA mediated by SCD [55].

### 4.4. Regulation of Triglyceride Synthesis by trans-10, cis-12 CLA

The mammary gland of ruminants is a highly efficient organ for the production of triacylglycerol (TAG). The process of TAG synthesis involves the progressive addition of fatty acyl groups to glycerol-3-phosphate, with the initial step being catalyzed by mitochondrial glycerol-3-phosphate acyl transferase (GPAM) [56]. The intermediate step of phosphatidic acid (PA) formation is facilitated by 1-acylglycerol-3-phosphate O-acyl transferase (AGPAT) enzymes, which esterify the sn-2 position of glycerol-3-phosphate. Among various isoforms in the bovine mammary gland, 1-acylglycerol-3-phosphate O-acyl transferase 6 (AGPAT6), located on Bos taurus autosome 27, is the most abundant. Another gene, phosphatidic acid phosphohydrolase1(LPIN1), plays a crucial rule in catalyzing PA dephosphorylation to form diacylglycerol (DAG) that is involved in the synthesis of TAG in BMECs [57,58]. Subsequently, Diacylgycerolacyltransferase (DGAT) and fatty acid acyl coenzyme A facilitate the formation of TAG [59]. Specifically, DGAT binds preferentially to the sn-3 position of DAG and undergoes acylation, where most of the butyric acid in milk TAG is located [60]. Similarly, the moderate upregulation of DGAT expression levels observed in early lactation confirms its pivotal role in TAG synthesis. Despite the inhibitory effect of high concentrations of *trans*-10, *cis*-12 CLA on triacylglycerol accumulation in BMEC, DGAT1 expression is detected to be upregulated [45]. This paradox stems from the fact that *trans*-10, *cis*-12 CLA’s inhibiting effect on the de novo synthesis of milk lipids is stronger than its promotion of the synthesis of triglycerides, with the final phenotype of reduced TAG accumulation in milk (Figure 4). Low doses of *trans*-10, *cis*-12 CLA, however, have no effect on triglyceride accumulation [61].

### 4.5. Regulation of Key Transcription Factors in Milk Fat Synthesis by trans-10, cis-12 CLA

#### 4.5.1. Regulation of SREBP1 by the *trans*-10, *cis*-12 CLA

The contribution of SREBP1 to fat synthesis through its mechanism for PUFA is widely recognized. During milk fat inhibition, results reveal a suppressed expression of related transcription factors, in addition to decreased levels of milk fat synthesis enzymes [62]. Via transcription factors involved in regulating milk fat synthesis, the action of CLA results in a synergistic reduction of lipogenic enzymes, particularly Sterol Regulatory Element Binding Transcription Factor 1 (SREBP1) and Peroxisome Proliferator-Activated Receptor Gamma (PPARG). These findings offer strong evidence that the principal mechanism responsible for the inhibition of milk fat caused by CLA is transcriptional regulation.

Furthermore, it is believed that increased milk fat synthesis in lactating cows and goats is linked to a high expression of SREBP1. This is because SREBP1 binds to sterol response elements in the promoter region of these genes, which regulates and activates the transcription of target genes primarily involved in lipogenesis [63]. SREBP1’s beneficial influence on the mTOR signaling pathway contributes to the decreases in triglyceride production and mRNA expression of ACC, FAS, SCD, and FABP3 observed in dairy cow mammary epithelial cells transfected with SREBP1 siRNA [64]. This effect has also been demonstrated in goats regarding milk lipid synthesis [65].

CLA exhibits inhibitory effects on the activity and expression of SREBP1, thereby downregulating the expression of genes related to milk fat synthesis. Specifically, *trans*-10, *cis*-12 CLA inhibits the expression of SCAP and INSIG1, which are responsible for the abundance of nuclear SREBP1 (nSREBP1), thereby suppressing the activation of SREBP1. Subsequently, mRNA levels of SREBP1 downstream genes ACACA, FASN, and SCD1 are all downregulated [66,67]. In vitro studies have shown that *trans*-10, *cis*-12 CLA treatment does not alter the transcription and translation levels of nSREBP1, but directly suppresses and reduces its mRNA levels when 26S protease activity is inhibited.

#### 4.5.2. Regulation of THRSP by *trans*-10, *cis*-12 CLA

Nuclear protein thyroid hormone response site 14 (THRSP, Spot14, S14) moderately expresses in the mammary gland and has a high expression in the liver and adipose tissue, where it controls the synthesis of milk lipids. In goat epithelial cells of the mammary gland, a high level of S14 expression results in upregulation of FASN, SCD1, and GPAM expression and a downregulation of CD36 expression but no change in ACACA expression [68]. Analysis of microarray junctions of bovine mammary tissue cultures reveals S14 as a candidate gene in response to *trans*-10, *cis*-12 CLA, the same as SREBP1. Moreover, cDNA microarray analysis shows that *trans*-10, *cis*-12 CLA-treated bovine mammary tissue exhibits a downregulation of S14 mRNA expression. Changes in protein levels are not examined. [69]. The S14 promoter has SRE at the individual nucleotide level, and nSREBP1 has a significant impact on how it expresses. Consequently, S14’s potential as a supplementary cellular signal or transcriptional coactivator of SREBP1 has been suggested in transgenic mice [70], but its applicability in ruminants is still up for debate.

#### 4.5.3. Regulation of PPARG by *trans*-10, *cis*-12 CLA

In mammary tissue, PPARG is another important transcription factor that regulates ruminant milk fat metabolism by facilitating the expression of FASN, ACC, LPIN1, and so on [71]. Notably, its activation is attributed to the C-terminal amino acid residues at 217-399 of LPIN1 [55]. Surprisingly, the PPARγ agonist TZD did not increase the amount of lipids produced by nursing dairy sheep or reverse the detrimental effects of *trans*-10, *cis*-12 CLA on the genes SREBP1, SCD1, and mTOR that are involved in the production of milk lipids [72]. In addition, *trans*-10, *cis*-12 CLA did not seem to activate PPARG expression in bovine mammary epithelial cells [69]. However, *trans*-10, *cis*-12 CLA does impact milk fat synthesis through PPARs, and basal activation of PPARG is found to cause CLA to block the transcription of genes involved in milk fat synthesis [57,73]. Moreover, PPARG is found to regulate INSIG, a gene involved in the activation of SERBP1 [74]. Therefore, we speculate that when PPARG is suppressed, the restrictive effect of *trans*-10, *cis*-12 CLA on SREBP expression may be reversed.

In conclusion, a great deal of research has been done on the regulatory mechanisms of CLA, namely *trans*-10, *cis*-12 CLA, on the transcription factor SREBP1, which is implicated in the synthesis of milk lipids. These mechanisms include the inhibition of upstream factors leading to the reduction of SREBP1 levels in the nucleus, as well as the direct negative regulation of SREBP1 transcript levels under specific conditions (Figure 5). However, the regulation of the other transcription factors, THISP and PPARG, in ruminant milk fat synthesis remains unclear. It has been revealed that the expression of PPARG and SREBP1 is modulated by S14, while S14 does not exhibit any correlation with the expression of genes associated with lipid synthesis downstream of these two transcription factors [71]. According to the authors’ hypothesis, tautomers exist in bovine mammary cells that act similarly to S14. Moreover, *trans*-10, *cis*-12 CLA downregulates their expressions in an opposite way to S14, which reverses the expression of genes associated with the synthesis of milk fat.

### 4.6. Regulation of Energy Metabolism of Milk Fat Synthesis by trans-10, cis-12 CLA

During the periparturient phase, cows often experience inadequate energy intake to meet their maintenance and milk synthesis needs, leading to a negative energy balance (NEB). This condition places a significant demand on the liver to generate glucose to drive milk production and alter milk fat concentrations and lipid distribution [4]. This phase is reliant on the body’s mobilization of fat and spans from late gestation to mid-lactation, with varying responses from fat depots to the changing physiological state.

Milk fat synthesis notoriously consumes a significant amount of energy. However, the addition of *trans*-10, *cis*-12 CLA may induce MFD, with subsequent allocation of stored energy to body storage, thereby regulating the NEB in early to mid-lactation. Association between MFD and real gastric infusion-induced energy redistribution with *trans*-10, *cis*-12 CLA suggests that the production of MFD induced by the dietary supplementation of calving cows with *trans*-10, *cis*-12 CLA can reduce energy requirements and modestly improve the negative energy balance during the first week of lactation. This process is accompanied by a reduction in milk fat content (increased citrate concentration), a reduction in fat mobilization, and an increase in lipid droplets in the subcutaneous adipose tissue. Label-free quantitative analysis of phosphorylation proteomics in subcutaneous and omental adipose tissue from CLA-supplemented cows demonstrates adipogenesis. This is evidenced by the phosphorylation on PHDA1-associated sites (S431) in mitochondria and elevated levels of ACSS2 and ACLY. These proteins mediate the expression of acetyl coenzyme Cytoplasmic acetyl coenzyme A, which triggers fatty acid synthesis.

Furthermore, activation of acetyl coenzyme-forming enzymes promotes increased activity of the fatty acid synthase ACACA and FASN (Figure 6). Cytoplasmic accumulation of acetyl CoA initiates fatty acid synthesis, while the activation of acetyl CoA forming enzymes promotes the activity of fatty acid synthase ACACA and FASN. GO analysis reveals an enrichment of the AMPK pathway, which plays a crucial role in regulating lipogenesis. Additionally, an elevation in phosphorylation levels is found to enhance the lipolytic activity of lipolytic proteins hormone-responsive lipase (LIPE) and adipose triglyceride lipase (PNPLA2) in adipose tissue [75].

The Endocannabinoid System (ECS) plays a role in regulating adipose tissue metabolism [76]. This is evidenced by an increase in the phosphorylation levels of enzymes diacylglycerol lipase alpha (DAGLA) and perilipin (PLIN) after *trans*-10, *cis*-12 CLA supplementation. Specifically, DAGLA participates in converting DAG into 2-arachidonoylglycerol (2-AG). In summary, simultaneous involvement of CLA in both fat decomposition and fat generation in adipose tissue may account for the decrease in milk fat. Therefore, using *trans*-10, *cis*-12 CLA to reduce milk fat production is a viable option for mitigating metabolic imbalances in perinatal cows.

## 5. Effect of *cis*-9, *trans*-11 CLA on Milk Fat Synthesis

Unlike *trans*-10, *cis*-12 CLA, which clearly inhibits milk lipid synthesis, *cis*-9, *trans*-11 CLA exhibits more ambiguous and limited regulatory effects on the mammary fatty acid metabolism. Recent studies have found that *cis*-9, *trans*-11 CLA mediates the upregulation of genes related to milk lipid synthesis and lipid droplet synthesis, specifically through upregulating nuclear factor erythroid2-related factor 2 (NrF2) and repairing mammary epithelial cell homeostasis [77]. In response to Lipopolysaccharide (LPS) attack, mammary epithelial cells show oxidative stress and inflammation, which in turn disrupt cellular homeostasis. It is believed that Nrf2, as a key to reversing oxidative stress, functions as an antioxidant response element (ARE) binding transcription factors in the promoters of antioxidant target genes such as superoxide dismutase (SOD) and heme oxygenase-1 (HMOX1) [57]. The administration of CLA to mammary epithelial cells stimulated by LPS significantly results in an increase in the ratio of phosphorylated Nrf2/total Nrf2 and the protein expression of downstream HMOX1, thereby reversing the reduction of BMEC by LPS stimulation [68]. These findings suggest that the Nrf2 pathway may serve as a crucial pathway for CLA to reduce oxidative stress.

In addition, LPS-induced oxidative stress can be alleviated by autophagy, an innate immunity pathway that defends against pathogenic bacteria and suppresses inflammatory responses [78]. Autophagy can be partially activated by excessive accumulation of reactive oxygen species (ROS) through the ROS-Nrf2-p62-autophagy signaling pathway. Results from immunoblot analysis demonstrate that prolonged oxidative stress increases the abundance of the LC3B protein, and lysosome formation decreases the expression of autophagic substrate p62, indicating an enhancement of autophagic flow. However, pretreatment with *cis*-9, *trans*-11 CLA results in downregulations of autophagy-related proteins LC3B, Beclin1, and ATG5 and increases in protein abundance and immunofluorescence intensity of p62, indicating the Nrf2 signaling pathway leading to a reduction in autophagy, thereby restoring cellular homeostasis (Figure 7). This process is accompanied by an upregulation of transcription factors SREBP and PPARG, and downstream expression of genes related to milk fat synthesis. Taken together, *cis*-9, *trans*-11 CLA attenuates the lipopolysaccharide-induced inhibition of fatty acid synthesis by inhibiting oxidative stress and autophagy in bovine mammary epithelial cells [79].

## 6. Discussion

CLA is a vital nutrient that is naturally present in both human and animal bodies. It is a beneficial fatty acid that cannot be synthesized in the human body and requires dietary intake, with dairy products being its main source. Notably, yak milk contains the highest amount of CLA, accounting for 0.92% of total FAs, while this proportion in regular cow milk ranges from 0.4% to 0.79% [80]. The biosynthesis of CLA involves both rumen hydrogenation and endogenous synthesis, with the latter being the main source. Studies have shown that diminished expression of LAI may cause low CLA production [36]. Despite attempts by researchers to improve enzyme activity through the integration of gene cloning with targeted mutagenesis and directed evolution, substantial progress has yet to be achieved [81]. Since LA does not undergo direct conversion to CLA in rumen but instead passes through the intermediate TVA, this step seems to reduce the efficiency of CLA production. Consequently, CRISRP/CAS9 technology, which enables precise genome editing, has been effectively employed in dairy cows. It may be a novel and interesting idea to use CRISRP/CAS9 technology to insert genes into the ruminant genome to block the conversion of LA to TVA and directly to CLA, thereby improving CLA. In addition, the utilization of probiotics in ruminants has been shown to improve milk production performance by altering rumen fermentation and microbiota [82]. In addition to probiotics, of course, there are other methods used to modulate biohydrogenation in the rumen to increase the availability of pentenoic acid in the rumen, and these will be important factors in improving the fat composition of ruminant products.

Numerous experiments have demonstrated that yeast strains exhibit beneficial effects on milk fat production. Specifically, they stimulate the production of acetate and cellulolytic bacteria, precursors of milk fat synthesis, while simultaneously elevating rumen pH to mitigate the growth of *trans*-10, *cis*-12 CLA, thus alleviating milk fat inhibition [83]. Meanwhile, rumen lysing cellulolytic bacteria produce more TVA, allowing part of *cis*-9, *trans*-11 CLA to escape biohydrogenation and be absorbed into milk fat [20]. Drawing from the evidence that *cis*-9, *trans*-11 CLA promotes milk fat synthesis, we hypothesize that this pathway potentially alters milk fat composition while increasing milk fat content. The addition of probiotic complexes in diets has also been shown to improve milk fat production [84], although there is a dearth of empirical data to support the notion that it modifies *trans*-10, *cis*-12 CLA-induced milk fat suppression or that it affects *cis*-9, *trans*-11 CLA content.

At present, the regulatory mechanism of *trans*-10, *cis*-12 CLA on milk fat synthesis genes and related transcription factors is well-established, but limited reports have been conducted on its effects on the signaling pathways of milk fat synthesis. Nevertheless, *trans*-10, *cis*-12 CLA has been shown to increase the activation of fat metabolism signaling pathway targets, such as the activation of the mTOR signaling pathway, which may offer insights into the study of milk fat metabolism signaling pathways [75]. Unlike *trans*-10, *cis*-12 CLA, which inhibits milk fat synthesis, *cis*-9, *trans*-11 CLA, another important isomer, is generally considered to promote milk fat synthesis. In addition, recent studies have revealed that CLA-induced milk fat inhibition alters the composition of milk fat globule membrane proteins, resulting in a reduction in milk fat globules. This also confirms the involvement of CLA in regulating lipid droplet formation [85].

In addition to dairy products, breast milk has been reported to contain trace amounts of CLA. Strains capable of producing CLA have been isolated and identified in breast milk, predominantly Bifidobacterium and *Lactobacillus*, which are also present in the feces of breastfed infants. Human milk oligosaccharides (HMO), exclusive to breast milk, have been demonstrated to stimulate the growth of these two strains. HMO, also referred to as the “bifidogenic factor,” can serve as a carbohydrate source for Bifidobacterium, leading to extensive proliferation [86]. Despite being unable to utilize HMO directly, *Lactobacillus* can utilize carbohydrates present within it. Furthermore, HMO is expected to indirectly regulate CLA production by selectively promoting bacterial growth and regulating intestinal flora, since certain bifidobacteria and lactobacilli have the ability to produce healthy CLA. This regulation is achieved by affecting the proliferation of high CLA-producing strains, improving energy metabolism and lipolysis, and becoming a potential mechanism for reducing the risk of obesity in infants.

The inhibitory effects of CLA on milk fat synthesis in humans are similar to those in ruminants, sharing some commonalities among specific mechanisms, including the inhibition of fatty acid synthesis and increasing lipolysis. Recent studies have indicated that CLA supplementation leads to a greater stimulatory effect of carnitine-palmitoil- transferase-1 (CPT-1) on lipolysis in beta-oxidation, as compared to the transport of fatty acid acyl coenzyme A complexes into the mitochondrial machinery [87]. CPT-1 is also known to be a joint regulator of energy metabolic processes in cows and is present in leukocytes and hepatocytes. Therefore, we speculate that there is great potential for CPT-1 to ameliorate the negative energy balance caused by CLA.

Admittedly, the inhibitory effect of *cis*-9, *trans*-11 CLA on milk fats has not yet been fully investigated. In practical production, it is imperative to consider different dietary collocations, combined with nutritional and feeding management, and other aspects, to reduce *trans*-10, *cis*-12 CLA production and improve milk fat rates. Presently, the application of CLA in ruminants primarily focuses on milk fat synthesis and energy balance. However, there is no convincing evidence on how to improve the milk fat rate by maintaining the homeostasis of rumen epithelial function. Therefore, further comprehensive research is essential to maintain the health of ruminants. In addition, *trans*-10, *cis*-12 CLA is of crucial value to human health and ruminant dairy products, serving as a pivotal source of human CLA. The authors posit that effective techniques and methods are indispensable to mitigate milk fat inhibition and retain CLA in milk, thereby meeting human demand for this beneficial nutrient. Obviously, numerous biological enigmas persist regarding the regulation of the milk fat metabolism by CLA.

## Figures and Tables

**Figure 1 animals-13-03429-f001:**
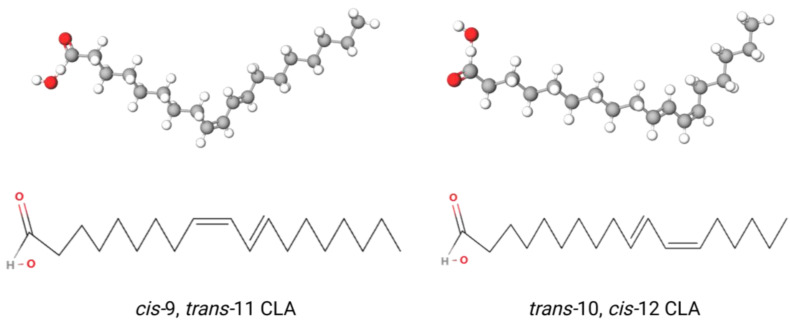
The structure of *cis*-9, *trans*-11 CLA and *trans*-10, *cis*-12 CLA. The red parts indicate the double bond between the oxygen and carbon atoms and the hydroxide bond between the oxygen and hydrogen atoms.

**Figure 2 animals-13-03429-f002:**
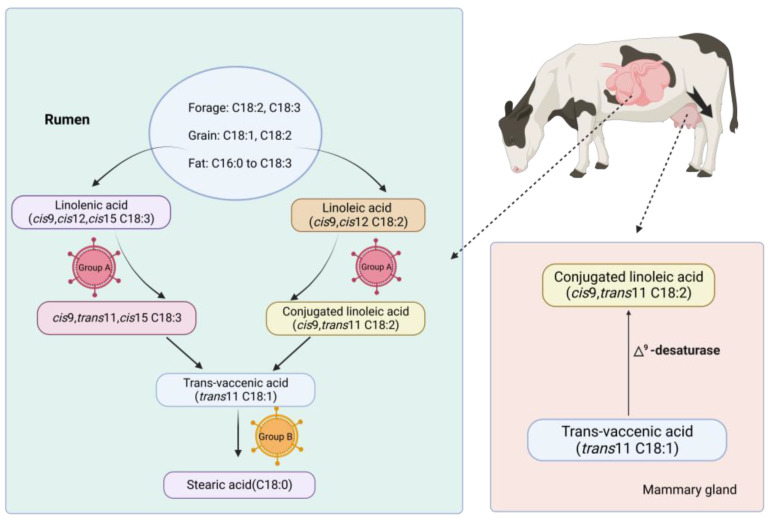
The biosynthesis process of CLA in ruminants (cattle). The biosynthesis of CLA in ruminant milk involves two primary pathways: endosynthesis and biohydrogenation. In the rumen, lipids undergo initial hydrolysis into LA and LNA, which are subsequently transformed through isomerization and hydrogenation into TVA by the isomerase enzyme secreted by group A bacteria, and stearic acid by group B bacteria. Notably, the products of isomerization differ between the two. Unlike biohydrogenation, the *cis*-9, *trans*-11 CLA formed in the rumen is subject to partial absorption or metabolic transformation by rumen microorganisms, resulting in the formation of TVA. The synthesis of *cis*-9, *trans*-11 CLA is heavily reliant on Δ^9^-Desaturase, which participates in the final step of endogenous synthesis of CLA via breast TVA.

**Figure 3 animals-13-03429-f003:**
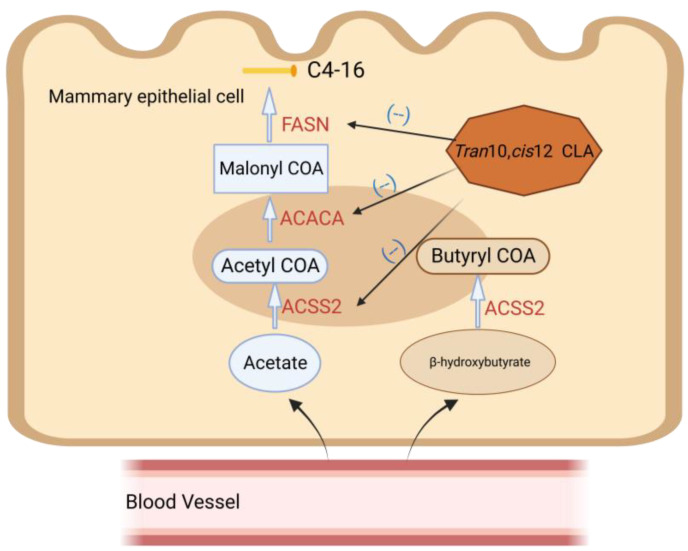
Regulation of CLA on de novo synthesis of milk fat. Fatty acid synthesis substrates include acetate and butyrate. After entering the mammary gland from plasma, they synthesize and extend the carbon chain of FAs under the action of ACSS2, ACACA, and FASN. *Trans*-10, *cis*-12 CLA inhibits de novo synthesis of milk fat, specifically by downregulating the expression of genes related to de novo synthesis such as ACSS2.

**Figure 4 animals-13-03429-f004:**
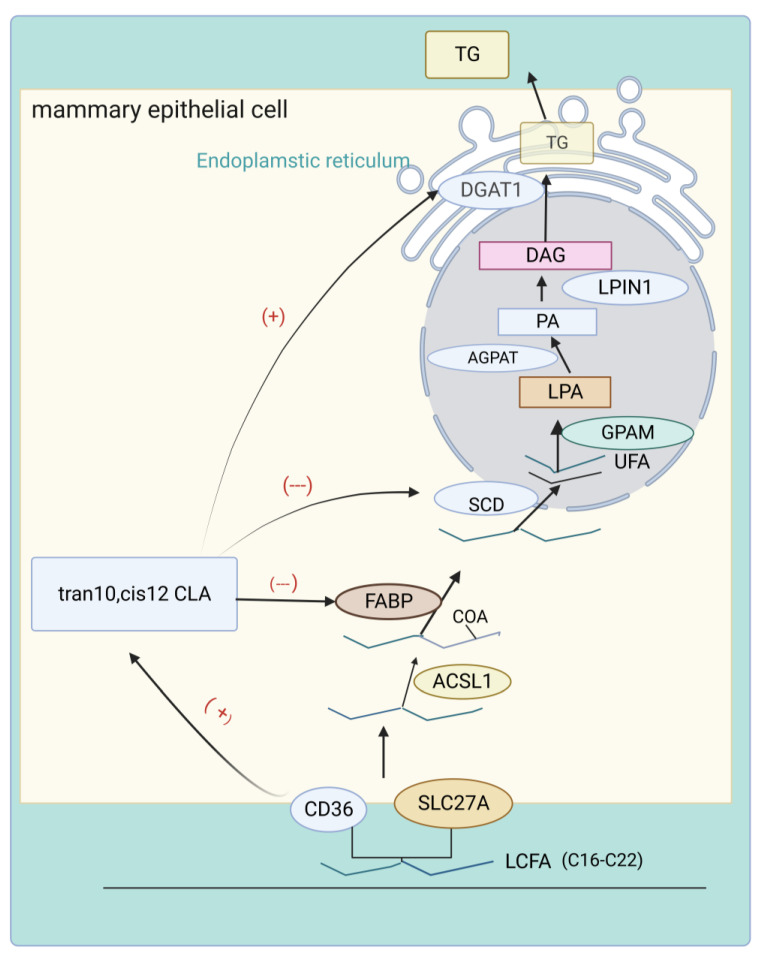
The regulatory mechanism of CLA on the uptake, transport, and triglyceride synthesis of LCFAs in milk fat. Almost all LCFAs in the milk of ruminants (including cows) originate from exogenous LCFA absorbed from the digestive tract. After entry into the bloodstream, they are transported across the membrane to the mammary epithelium, where they contribute to the synthesis of milk lipids. The fatty acid transport proteins involved in this process include CD36, SLC27A, ACSL1, and FABP3. The saturated FAs are desaturated by the enzyme SCD. Triglycerides are formed in the endoplasmic reticulum with regulations of GPAM, AGPAT, LPIN1, and DGAT1. Numerous studies have confirmed that high concentrations of *trans*-10, *cis*-12 CLA inhibit the expression of FABP and CD36 but play a significant role in promoting the expression of SCD and DGAT1.

**Figure 5 animals-13-03429-f005:**
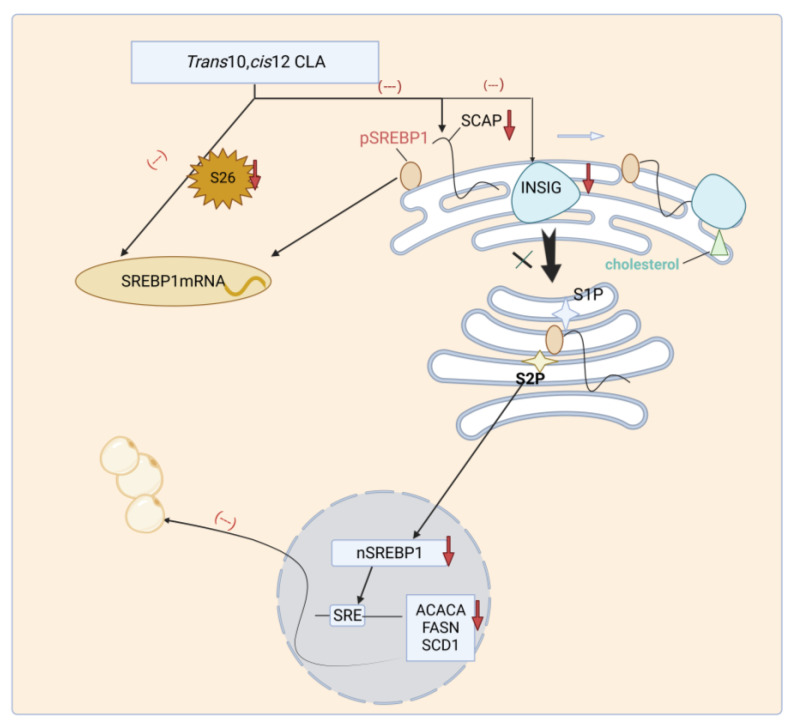
*Trans*-10, *cis*-12 CLA regulates the regulation mechanism of milk fat synthesis by acting on SREBP1. SREBPs, transcription factors involved in ruminant milk fat synthesis, exist as inactive precursor proteins. They need to be transported by SCAP from the endoplasmic reticulum to Golgi apparatus before being processed into active transcription factors, nSREBPs. This process is contingent upon the intracellular levels of insulin-induced gene (INSIG), which encodes protein site 1 proteolytic enzyme (S1P) and site 2 proteolytic enzyme (S2P). On the one hand, *trans*-10, *cis*-12 CLA inhibits the expression of SCAP and INSIG1, thereby hindering the activation of SREBP1 and reducing the transcription level of genes related to milk fat synthesis downstream of SREBP1. On the other hand, only when 26S protease activity is inhibited does SREBP1 mRNA undergo direct negative regulation by *trans*-10, *cis*-12 CLA.

**Figure 6 animals-13-03429-f006:**
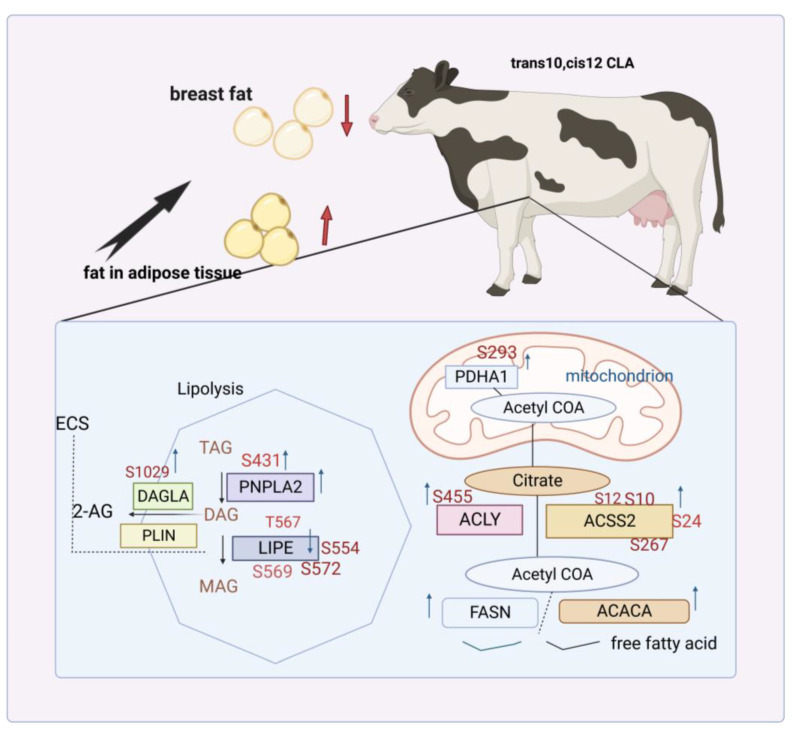
Mechanism of supplementing *trans*-10, *cis*-12 CLA to alleviate negative energy balance in early lactation. Upregulation of phosphorylation at PHDA1-related sites in mitochondria is detected in the subcutaneous and omental adipose tissue of cows supplemented with *trans*-10, *cis*-12 CLA, indicating the production of mitochondrial coenzyme A. Increased phosphorylation levels of ACLY and ACSS2 in the cytoplasm promotes the expression of acetyl CoA in the cytoplasm and upregulates levels of fat synthesis-related genes ACACA and FASN. In addition to fat synthesis, upregulation of PNPLA2 is related to lipolysis at serine 431 site and downregulation of the enzyme LIPE. The ECS involved in regulating lipid metabolism is also affected by CLA, further verifying that decreased milk fat energy output increases fat generation.

**Figure 7 animals-13-03429-f007:**
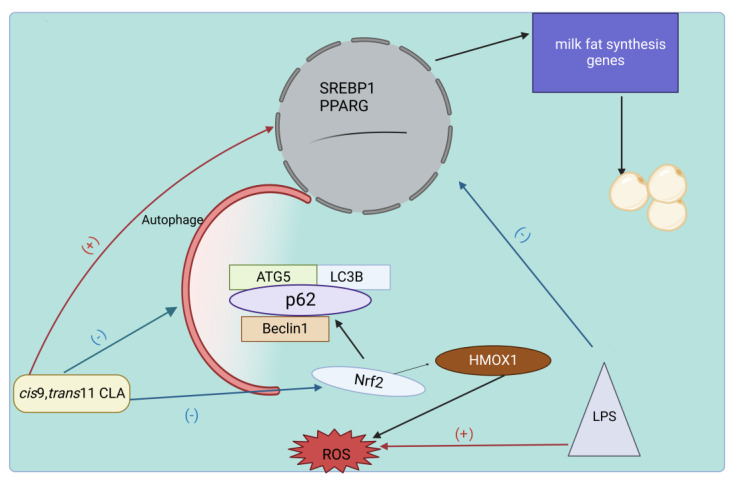
The regulatory mechanism of *cis*-9, *trans*-11 CLA on milk fat synthesis. Once invaded by LPS, the homeostasis of mammary gland epithelial cells is disrupted, thereby affecting milk fat synthesis. *Cis*-9, *trans*-11 CLA alleviates oxidative stress by inhibiting the transcription factor Nrf2 and its target gene HMOX1 pathway. Further validation is conducted to weaken the occurrence of autophagy through the ROS-Nrf2-p62 autophagy signaling pathway. This process is accompanied by upregulation of transcription factors SREBP and PPARG, and gene expression related to downstream milk fat synthesis.

## Data Availability

No new data were created or analyzed in this study. Data sharing is not applicable to this article.

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
