# Peer review of "Progress of Conjugated Linoleic Acid on Milk Fat Metabolism in Ruminants and Humans"

_animals, 2023, doi:10.3390/ani13213429_

Round 1

Reviewer 1 Report

Comments and Suggestions for Authors

The review is well structured, with several topics from the CLA synthesis to the effect of each two CLA isomers on the synthesis of milk fat.

The title can induce in some error, since the reader can expect an review from studies in milk fat of ruminants, but along of the manuscript are several references to human milk. The authors can improve the title considering this.

There are several inconsistencies in the writing style, which is not uniform throughout the manuscript, namely:

- cis and trans should be in italics.

- the authors need to uniformize the presentation form of fatty acids, (cis-9, trans-11) in some cases are presented cis9, trans11 or c9, t11. Please correct along of the manuscript even in the figures. 

- the species name of microorganisms should be in italic, as the in vitro and in vivo, and de novo.

- There are several abbreviations there appear before or even without their meaning. Please confirm along of the manuscript.

- In Figures, can add adapted from...

Specific comments:

Ln 13: add (CLA).

Ln 16: CLA instead "conjugated linoleic acid".

Ln 17: cis-9, trans-11 18:2. The authors can use these form or cis-9, trans-11 CLA, but not only "cis-9, trans-11", please review and change along of the manuscript.

Ln 18: isomers instead "ones".

Ln 25: ruminant instead "ruminous".

Ln 52 and 58: add a reference.

Ln 53: CLA isomer.

Ln 67: Not only these isomers, as the authors referred on Ln 71. Please rewrite the sentence.

Ln 70: cis-cis.

Ln 78: the figure can be improved.

Ln 92: Butyrivibrio ... Clostridium. The C. proteoclasticum was renamed later as B. proteoclasticus, please see Moon et al. 2008,  International Journal of Systematic and Evolutionary Microbiology 58.9 (2008): 2041-2045.

Ln 96-97: cis-9, cis-12 18:2 and cis-9, cis-12, cis-15 18:3.

Ln 99: cis-9, cis-12 18:2

Ln 100: trans-11 18:1

Ln 105: trans-11

Ln117-120: The endogenous synthesis via delta-9 desaturase represents the predominant source of CLA in ruminant tissues. About 80% of c9,t11CLA deposited in tissues is originated by endogenous desaturation of TVA produced in rumen (Palmquist et al. 2004)

Ln 132: Please correct the figure according to the uniformization of the fatty acids.

Ln 159-160: add a reference.

Ln 211, 227, 229: add a reference. 

Ln 224: mammary gland instead "breast" (is referred to ruminant not human, please change along of the manuscript).

Ln 228: FASN.

Ln 290-293: add a reference.

Ln 313: add a reference. qPCR

Ln 320-323, 444-447, 480, 489, 491, 493: add a reference.

Ln 474: only true for human, please clarify.

Ln 482-484: Please clarify, because in rumen this occurs, only in tissues  endogenous synthesis is required. 

Ln 485-489: Others than probiotics are also used to modulate the ruminal biohydrogenation to increase the vacenic acid availability in the rumen, which is crucial for the endogenous synthesis of CLA in tissues, and improve the fat composition of ruminant products.

Ln 541-547: Instead trans-10, cis-12 18:2 the cis-9, trans-11 18:2 is of crucial value for both human and animal health, mainly due to their beneficial biological activities. Furthermore, trans-10, cis-12 18:2 can originate trans-10 18:1 in rumen, which is not a precursor of cis-9, trans-11 18:2, so there is no endogenous synthesis from trans-10 18:1 to CLA. And trans-10 18:1 has also been associated with MFD syndrome with potential adverse health effects. Then, these sentence can induce in error, please clarify and rewrite the sentence.

Author Response

Response to Reviewer 1 Comments

Point 1: The title can induce in some error, since the reader can expect an review from studies in milk fat of ruminants, but along of the manuscript are several references to human milk. The authors can improve the title considering this.

Response 1: Thanks to your suggestion, we have appropriately modified our title to ensure that it corresponds to the content of the manuscript.

Point 2: There are several inconsistencies in the writing style, which is not uniform throughout the manuscript…

Response 2: Thanks to your suggestion, we have have standardised the representation of fatty acids as cis-9, trans-11 and have made corrections throughout the manuscript, including figures. We have amended the nouns that need to be italicised and given the full names of all abbreviations, etc. ...

Point 3: Specific comments: …

Response 3: We have revised the manuscript point by point in response to your comments and can visible in the revised manuscript. We are confident that our manuscript will meet your requirements. Thank you again.

Reviewer 2 Report

Comments and Suggestions for Authors

The current submitted manuscript by Wang et al is good on “Research progress in conjugated linoleic acid on milk fat me- metabolism in ruminants.” I have reviewed the complete manuscript, and the following points should be taken as suggestions.

L39 milk fat “rate” change to “composition”

L87 suggests providing some figure-related process.

L133 1. Spell out all abbreviations used in Figure 2 and other Figures.

L142 “infants,” did you mean in the ruminant digestive tract? Please. rephase to be clarified.

L157 provides more information on Lactobacillus in the digestive tract of ruminants.

L174 provides information on Lactobacillus in the digestive tract of ruminants.

L216 provides a related Figure.

L254 Spell out all abbreviations used in Figures 4, 5, 6.

L515 Lactose on one type of disaccharide that is available in milk, not clear that oligosaccharides available in milk composition, specify type and details.

Author Response

Response to Reviewer 2 Comments

Point : L39 milk fat “rate” change to “composition”…

Response : Thank you for your suggestions, we have revised the content of the manuscript in its entirety based on the relevant comments, all acronyms have been explained in the manuscript or diagrams, and all images have been improved. The wrong content in the manuscript has been corrected and can visible in the revised manuscript. We believe that our manuscript can meet the requirements and express our gratitude.

Reviewer 3 Report

Comments and Suggestions for Authors

General remarks:

The topic of this manuscript is very interesting for Animals’ readers, I find only some problematic parts in the text.

Detailed opinions:

line 39: please use „…cows’ milk…” instead of „…cows ’milk…”!

line 43: short vs medium, in this line short is C4-C8, but the medium also starts C8! May better, if use C9-C14 as medium-chain FAs!

Lines 44-45: please correct this sentence, because C16:0 is partially (~50%) is synthesis de novo in the mammary gland, the other part originated from the intestine!

lines 59-60: please add that which CLA isomer has a positive effect on human health!

line 70: “…cis-cis-cis…”?

Figure 3: I think this figure is connected to 4.1 not the 4.2! 4.2 rather is connected to Figure 4!

line 271: trans10!

line 279: machine? Please think about it!

lines 358-361 (+394 and other places): please add the accurate isomer phrase to the CLA!

Discussion section: add the exact CLA isomer(s) to the text!

Please add the Conclusions section to the manuscript!

Author Response

Response to Reviewer 3 Comments

Point : Detailed opinions…

Response : Thank you for your suggestions, we have made appropriate changes to the content of the manuscript based on the relevant comments, the wrong expression in the sentence has been corrected. The content and pictures of the manuscript have been improved and can visible in the revised manuscript. We believe our manuscript meets the requirements and express our gratitude.
